# Mechanotransduction in Mesenchymal Stem Cells (MSCs) Differentiation: A Review

**DOI:** 10.3390/ijms23094580

**Published:** 2022-04-21

**Authors:** Narmadaa Raman, Siti A. M. Imran, Khairul Bariah Ahmad Amin Noordin, Wan Safwani Wan Kamarul Zaman, Fazlina Nordin

**Affiliations:** 1Centre for Tissue Engineering and Regenerative Medicine (CTERM), Faculty of Medicine, Universiti Kebangsaan Malaysia, Jalan Yaacob Latiff, Bandar Tun Razak, Cheras, Kuala Lumpur 56000, Malaysia; raxlenar25@gmail.com (N.R.); siti.imran@ukm.edu.my (S.A.M.I.); 2Department of Microbiology, Faculty of Science, Universiti Tunku Abdul Rahman, Kampar 31900, Malaysia; 3School of Dental Sciences, Universiti Sains Malaysia, Kampus Kesihatan Kubang Kerian, Kubang Kerian 16150, Malaysia; kbariah@usm.my; 4Department of Biomedical Engineering, Faculty of Engineering, Universiti Malaya, Kuala Lumpur 50603, Malaysia; wansafwani@um.edu.my

**Keywords:** mechanotransduction, differentiation of mesenchymal stem cells (MSCs), mechanical forces, mesenchymal stem cells (MSCs) therapy, regenerative medicine, tissue engineering

## Abstract

Mechanotransduction is the process by which physical force is converted into a biochemical signal that is used in development and physiology; meanwhile, it is intended for the ability of cells to sense and respond to mechanical forces by activating intracellular signals transduction pathways and the relative phenotypic adaptation. It encompasses the role of mechanical stimuli for developmental, morphological characteristics, and biological processes in different organs; the response of cells to mechanically induced force is now also emerging as a major determinant of disease. Due to fluid shear stress caused by blood flowing tangentially across the lumen surface, cells of the cardiovascular system are typically exposed to a variety of mechanotransduction. In the body, tissues are continuously exposed to physical forces ranging from compression to strain, which is caused by fluid pressure and compressive forces. Only lately, though, has the importance of how forces shape stem cell differentiation into lineage-committed cells and how mechanical forces can cause or exacerbate disease besides organizing cells into tissues been acknowledged. Mesenchymal stem cells (MSCs) are potent mediators of cardiac repair which can secret a large array of soluble factors that have been shown to play a huge role in tissue repair. Differentiation of MSCs is required to regulate mechanical factors such as fluid shear stress, mechanical strain, and the rigidity of the extracellular matrix through various signaling pathways for their use in regenerative medicine. In the present review, we highlighted mechanical influences on the differentiation of MSCs and the general factors involved in MSCs differentiation. The purpose of this study is to demonstrate the progress that has been achieved in understanding how MSCs perceive and react to their mechanical environment, as well as to highlight areas where more research has been performed in previous studies to fill in the gaps.

## 1. Introduction

In the human body, living cells and tissues are affected by physical forces and chemical stimuli. Mechanotransduction is defined as a force-induced process that initiates biochemical reactions such as modifying binding affinity, altering phosphorylation state, altering conformation, starting signaling pathways leading to gene expression, protein synthesis, and cellular phenotypic modification [1]. The process of translating physical forces to biochemical signals and translating these signals for cellular responses, mechanotransduction, thus regulates the differentiation of MSCs. Mesenchymal somatic or stem cells have self-renewal properties and can differentiate into osteoblasts, adipocytes, chondrocytes, tenocytes, cardiomyocytes, keratinocytes, hepatocytes, and neural cells [2]. MSCs possess a promising cell source for tissue engineering applications for treating injured tissue and many deteriorating diseases [3]. In addition to inhibiting tissue injury, MSCs show immunomodulatory properties, including anti-inflammatory and immunosuppressive abilities, which subsequently promote angiogenesis and stimulate mitosis of tissue-intrinsic stem cells.

Despite being isolated from the bone marrow, MSCs follow the minimum requirements set by the International Society for Cellular Therapy (ISCT), whereby MSCs should adhere to plastic under standard culture conditions; test positive for cell surface markers CD105, CD73, and CD90 and lack expression for CD11b or CD14, CD19 or CD79a, CD34, CD45, and HLA-DR, with the capability to differentiate into osteoblasts, adipocytes, and chondroblasts in vitro [2]. MSCs administered to heart disease improve cardiac function and inhibit scar due to stimulation of endogenous repair mechanisms such as regulation of immune responses, tissue perfusion, inhibition of fibrosis, and proliferation of resident cardiac cells [4]. Novel cell-based therapeutic techniques are required to achieve complete cardiac recovery [5].

Many studies revealed that MSCs therapy, including cell combinations, the inclusion of MSCs into biomaterials, and genetic modification of MSCs increases their release of paracrine substances, such as exosomes, growth factors, and microRNAs, have the potential to improve treatment effectiveness [6]. Further research into the appropriate method of administration, such as the proper dosage, the best cell populations, and treatment time, is required in regenerative medicine of cardiovascular disease [7]. Moreover, the efficacy of Mesenchymal stem cell therapy, on the other hand, is based on the determination of desirable phenotypes.

Uncontrolled differentiation can result in the unwanted differentiation of an undesirable phenotype, which can be harmful. Previous studies revealed that unwanted differentiation of bone marrow-derived MSCs (BMSCs) might cause intramyocardial calcification, and liver injury and induce vascular calcification after balloon angioplasty in hyperlipidemia rats [8]. These can be prevented, according to researchers, through practicing proper procedures in directing the differentiation process in MSCs consisting of various stimuli. Most biochemical stimuli, for example, small molecules, growth factors, and microRNA are important for cell surface receptors interaction, additionally, it is useful for the differentiation process and biophysical forces involving cyclic mechanical strain, fluid shear stress, matrix stiffness, and topography, microgravity, and electrical stimulation, which determines the shape and size of MSCs [9].

## 2. MSCs Properties

Adult or multipotent stem cells and pluripotent stem cells are derived from stem cells, whereby embryonic stem cells are well known as pluripotent stem cells, while mesenchymal stem cells, prominent as multipotent stem cells, are shown in Figure 1, were characterized according to their importance. The stem cells were demonstrated to have the capacity to develop into embryonic structures, while adult stem cells are multipotent stem cells capable of undergoing multiple differentiation with self-renewal potential [10]. Friedenstein identified a novel form of stromal cells in human bone marrow in 1970, which later were well recognized as MSCs. MSCs are found in a broad variety of tissues other than bone marrow, including adipose tissue, lung tissue, synovial membrane, endometrium, and peripheral blood [2]. Currently, the majority of MSCs employed in clinical trials for human benefits originate from bone marrow, adipose tissue, umbilical cord, and cord blood [10]. Bone marrow and adipose tissue are the primary sources of MSCs, yet the complexity of isolating MSCs from umbilical cord blood has been extensively investigated. While MSCs derived from various sources display distinct immunophenotypes, differentiation potential, transcriptome, proteome, and immunomodulatory activity, there is still much that is not fully understood.

MSCs generated from bone marrow and adipose tissue have been extensively investigated. Both types of MSCs have been extracted in large numbers, which possess immunomodulatory properties, and have a high capacity for extracellular matrix production even though they are not similar. MSCs originated from adipose tissue (AT-MSCs) synthesizing collagen (I, II, and III), but MSCs obtained from bone marrow (BM-MSCs) have a stronger proangiogenic activity and immunosuppressive impact than AT-MSCs [11]. BM-MSCs are the most competent cell type for suppressing induced T cell proliferation, and they demonstrated a strong anti-proliferative impact on phytohemagglutinin-primed T cells through a paracrine mechanism [12]. MSCs generated from induced pluripotent stem cells (iPSC-MSCs) have similar physical characteristics to BM-MSCs with multi-lineage potential to differentiate into various tissues of mesodermal origin and induce a regenerative microenvironment; besides, they can heal tissue in ischemic illness and have significant anti-inflammatory properties [13]. iPSC-MSCs proliferate more rapidly than BM-MSCs and can be grown for up to 40 passages while retaining a normal diploid karyotype and a consistent gene expression and surface antigen profile [14]. In addition to these intrinsic benefits, iPSC-MSCs play an important role in ischemic tissue healing.

Other possible sources of MSCs include the umbilical cord matrix, amniotic fluid, peripheral blood, and the heart; these types of MSC seem to possess unique features. While MSCs isolated from the umbilical cord matrix (UCM-MSCs) reduce T-cell activation similarly to BM- and AT-MSCs, B-cells exhibit no obvious changes, in contrast to BM- and AT-MSCs, which both inhibit B-cell activation [15]. MSCs produced from amniotic fluid (AF-MSCs) have multi-lineage potential, in contrast to BM-MSCs, which may additionally express the key pluripotency transcription factor Oct4 [16]. Furthermore, MSCs seen in peripheral blood are suggested to express pluripotency markers such as SSEA-4, Oct-3/4, and TRA-1-81. Although several studies reported the function of these markers is unknown due to these circulating cells lacking the ability to differentiate into pluripotent cells [14].

## 3. The Function of MSCs

The MSCs have several different roles in the body and cellular environment. The cellular phenotypes of MSCs change in different conditions, making this unique characteristic of MSC to support the maintenance of other cells. It was found that Grégoire et al. [17] identified three types of MSC derived from adipose tissue (AT), bone marrow (BM), and perinatal tissue (PT) derived tissues in an acute graft against a host disease mouse model (GvHD). Preclinical toxicity evaluation of clinical-grade placenta-derived decidual stromal cells (DSCs) in different preclinical models has been presented by Sadeghi et al. [18]. According to the International Society for Cellular Therapy (ISCT) in 2016, the MSC assay showed significant progress in predicting the therapeutic effect of MSCs. For the treatment of Bronchopulmonary dysplasia (BPD), MSCs are the most often used therapeutic agent due to their established safety profile, immune-evasion potential, relative ease of isolation, anti-inflammatory profile, and immunomodulatory activity [19].

Using AT-MSCs in fibrin glue, Masgutov et al. [20] reported promising preclinical outcomes on the regeneration of peripheral nerves. Such findings could subsequently increase the capabilities of endogenous MSC to encourage immunomodulatory and regenerative effects. To investigate this method, Ross et al. [21] used an anti-inflammatory, extremely low frequency pulsed electromagnetic field (PEMF) to treat rheumatoid arthritis. For more than two decades, MSCs have been used in research and clinical settings to show that they are safe for use as a therapeutic agent, especially in adults. There have been no human cases reported relative to tumorigenicity resulting from the administration of MSCs, which means that the risks of MSC treatment are quite low [19]. Furthermore, Soria-Juan et al. [19] elaborated on treatment involving critical limb ischemia and diabetes with cell products, specifically AT-derived MSCs, and their optimal delivery.

Moll et al. [22], on the other hand, have provided their findings based on allosensitization after local administration of allogeneic AT derived MSCs (Darvadstrocel, well known as Cx601, from Takeda/TiGenix) with detailed mechanistic studies on the complement system’s protection and attack susceptibility. Graft-versus-host disease was approved in Canada and New Zealand in 2012 for pharmaceutical use of MSCs in children to treat a 9-year-old boy with the illness after 10 years of investigation [23]. Recipients of allogeneic cell treatment may be at risk for immunological rejection; however, MSC transplantation reduces the chance of immunological rejection. MSC’s immune-evasive characteristics need to be better understood to develop MSCs as therapeutic agents. Some studies have shown that the immune system is underrated if Class I and II major histocompatibility complex (MHC) molecules are expressed at lower levels [15]. Additionally, MSCs have been isolated painlessly and ethically from the umbilical cord used to cure neonatal diseases. The MScs obtained from the umbilical cord may have improved proliferation and healing characteristics [19]. Bhavsar et al. [24], on the other hand, used direct current electrical stimulation (EStim) to stimulate mesenchymal stem cell proliferation and differentiation and concurrently measured the accompanying changes in mesenchymal stem cell resting membrane potential (RMP), which is crucial for a variety of inter-and intracellular processes that govern cell functions such as proliferation, differentiation, and migration, all of which are necessary for embryogenesis, repair, and regeneration. Thus it has been revealed that cell growth and differentiation can be sped up both chemically with pinacidil and diazoxide while physically involving DC EStim in the present experiments, which indicates this relationship is strong and consistent [24]. The overall MSC function shown in Figure 2 was demonstrated to express the potential of MSCs to be developed into regenerative medicine and tissues. 

## 4. Mechanotransduction in MSCs Differentiation

Mechanotransduction is the process by which the natural environment of a cell experiences a physical force that is exerted intracellularly and extracellularly, whereby later, the force is converted into biochemical and electrical signals resulting in cellular responses [25]. The revolutionized multiple molecular pathways from various discoveries and elucidation led to the basic and clinical understanding of the formation and development of tissues and organs. During the growth and emergence of the tissues, there are several involvements of mechanosignaling pathways such as interleukin pathways, transforming growth factor (TGF)-β [26], integrin, MAPK and G protein, β-catenin, tumor necrosis factor (TNF)-α and calcium ion pathways [27]. Extracellular matrix (ECM) and cellular mechanosignaling pathways are actively interconnected in tissue development. These have been demonstrated through MSCs and their response towards intercellular and extracellular signaling, which is transmitted in nuclear to alter protein activity and gene expression while extracellular signaling interrupts cells surrounding the matrix [28]; thus, the cell membrane and numerous intracellular components induce both signaling pathways, which have been shown to influence MSC differentiation as indicated in Figure 3 [29]. Meanwhile, the overall understanding of these interactions will be greatly advantageous in tissue engineering and pharmacological interventions that could facilitate the formation and development of tissues and organs.

### 4.1. Cell Membrane and Intracellular Components

Numerous cell membrane and intracellular components have been implicated in mechanotransduction, but this review will focus on primary cilia, extracellular matrix (ECM), Focal adhesion (FA), cytoskeleton, cadherins, integrins, and stretch-activated ion channels (SACs) as shown in Figure 3a.

#### 4.1.1. Primary Cilia

Most cells in the human body have membrane-encased microtubules that are well known as primary cilia. Primary cilia have been demonstrated as multifunctional antennas, which are considered to detect both chemical and mechanical cues from the internal as well as external environment [30]. The precise functions of primary cilia in mechanotransduction remain unknown because of their dual function as chemosensors and mechanosensors; thus, understanding how MSCs sense and respond to these signals is currently a highly researched area. MSCs could take such ‘outside-in’ and ‘outside-inside-out’ signals; therefore, MSCs differentiation requires more study to dissociate primary cilia’s chemo and mechanical sensing capabilities [31].

#### 4.1.2. Extracellular Matrix

The cell membrane plays a vital role in force transmission to the cell, where direct contact with the ECM occurs. Research on the impact of ECM on cell activities has been extensively studied over the past centuries. Cell adherence, shape, and migration, as well as the activation of signal transduction pathways controlling gene expression and dictating proliferation and stem cell destiny, were established by ECM itself [31]. ECM is composed of a structural macromolecular network which is meant to provide the appropriate structure and environment for delivered cells to take effect [1]. Components of the ECM are made up of solid collagen, laminin, GAGs (hyaluronic acid), chondroitin, and heparin, as well as soluble components such as metalloproteinases that mediate between ECM components [9]. Basement membranes and connective tissues can be distinguished by their composition and structural organization, which have been attributed primarily to ECM structures. The basement membranes provide two-dimensional support for cells originating from laminin and collagen IV, while connective tissues provide a fibrous three-dimensional scaffold to the cells that are mainly composed of fibrillar collagens, PGs, and GAGs [2].

The effects of ECM on cell functions have been extensively studied. Nowadays, it is well known that basement membranes and connective tissues influence ECM’s topography, viscosity, and mechanical characteristics delivered through elastic fibers, fibrillar collagens, GAGs, and related PGs, whereas collagens and elastin offer tensile strength [32]. ECM is composed of hard material, which creates deformation with the aid of a higher level of stress or soft material that is readily deformed at low stresses [31]. An interesting correlation has been found between stem cell differentiation and mechanical forces being exerted toward certain lineages resulting in the determination of ECM on cell fate [2]. When the ECM’s composition is altered, it may take on a distinctive geometrical shape that provides topographic and mechanical stimuli, both of which are essential for altering the phenotype of stem cells [1].

The dynamic crosstalk between the stem cell and ECM probably seems to alter the shape and ECM composition to secrete ECM structural components and matrix metalloproteinases besides sending mechanical signals via the cytoskeleton fibers. Overall, the idea is to create a favorable niche that induces mechanotransduction signals and guides stem cell fate [29]. Previous studies revealed that the alteration of ECM may lead to several consequences, such as affecting the biochemical and physical properties of ECM and leading to the abnormal organization of the network, which subsequently causes organ failure in humans. Cancer and fibrosis are caused by ECM’s disorganized composition, altering its mechanical properties [31]. In addition, Williams and Marfan syndromes occur due to the mutated gene encoding for elastin [25].

#### 4.1.3. Focal Adhesion

To respond to the external mechanical stimuli, the focal adhesion kinase (FAK) undergoes autophosphorylation, leading to intracellular mechanotransduction, which transduces downstream mechanotransduction [33]. The downstream stimulus involves the contraction of cytoskeletal cells and cell spreading, which develops FAK activation through FAK phosphorylation, which may be boosted by stretching or resistance through a stiff substrate [34]. FAK and the contractile cytoskeletal network develop tension in the cell to induce force into the nucleus [35]. Focal adhesions are mainly derived from the assistance of the integrin and intracellular environment, forming many signaling pathways, which are dependent on the actin cytoskeleton [34]. Many structural proteins, including b-subunits of integrins, vinculin, talin, and the actin cytoskeleton, are involved in their formation [31]. These formations assist integrin binding, other than determining MSCs differentiation and regulating cell shape. In addition, it contributes to signaling cascades to bind and convey signals from the cell to the nucleus.

Furthermore, the assembly of Focal adhesion provides activation of MAPK, GTPases, intracellular calcium concentration, FAK, and paxillin, together with their downstream signals, regulating MSC differentiation [33]. Extrinsic mechanical stimuli are known to induce the activity of Focal adhesions mechanically and their development and have a significant impact on mechanotransduction and MSC differentiation. Focal adhesions have been shown to increase downstream signaling cascades via cell-cell junctions and Focal adhesions interaction, which later generates complex signal production [32], on the other hand, focal adhesion proteins also serve as cytoskeleton anchor sites for actin and integrin filaments [33]. During the directional migration involving cell polarization and nucleus deformation, it has been shown that the migration results in nucleus squeezing and cytoskeleton local reorganization via FAK activation [27,35]. According to molecular dynamics studies, the FAK sensor is homeostatic and can auto-adjust to match the stiffness of the substrate [34].

#### 4.1.4. Cytoskeleton

The effects of cytoskeletons on cell functions have been extensively studied. Nowadays, it is well known that cytoskeletons control isometric tension in muscle via the actomyosin filament sliding mechanism [35]. The cytoskeleton is made of filaments that may control shape as well as deliver support to the cell [36]. The tension in cells allows the presence of a cytoskeleton at the binding site of integrin to transfer mechanical signals across cells [37]. The cytoskeletal tension is induced by intracellular signals, substrate stiffness, ligand type, and density, which in turn can impact focal adhesion interaction of the cell-cell junction process and regulate cell shape, which, as previously discussed, is a key regulator of MSCs differentiation [28].

Focal adhesion proteins also serve as cytoskeleton anchor sites for actin and integrin-binding [34]. RhoA is a key regulator of focal adhesions and stress fibers via downstream phosphorylation cascades that impact contractility and osteogenic-adipogenic determination, while Rho-associated protein kinase (ROCK) is a Rho effector involved in myosin contraction [38]. By manipulating Sox9 expression, RhoA/ROCK signaling has been demonstrated to encourage and hinder chondrogenesis. More often, the nucleus acts as mechanosensors, whereby the deformation of the nucleus in MSCs more often involves fibroblast nuclei via external and internal stimuli in terms of having identical cytoskeleton morphology. Additionally, it can disrupt chromatin morphology that eventually affects transcription. LINC, the linker of the nucleus and cytoskeleton complex, occurs through the actin and microtubules interaction with the aid of binding proteins, which are involved in DNA replication, gene expression, and transcription that influence MSCs differentiation [28].

#### 4.1.5. Integrins

Integrins are surrounded by ECM components, which lead to the interconnection of extracellular and cell internal environment to further establish linkage through integrins in the plasma membrane; thus, integrins can determine specific ligand binding due to the α and β subunits present in integrins [39]. It has been found that ECM sends the mechanical signal to integrin to activate signaling cascades involving macro-protein complexes [40], showing that integrin has less control over the cell characteristic compared to ECM [41]. Interestingly, the heterodimeric receptor integrins are shown to be involved in cardiac mechanotransduction through the binding of their extracellular environment to cardiomyocytes [40]. The integrins exhibit 24 types of integrin receptors which are non-covalently bonded to α and β-subunits, which consist of eighteen different α-subunits and eight different β-subunits. The α-subunit integrins induce ECM ligand specificity compared to β-subunits deliver cytoplasmic intracellular signals and later connect the sarcomeres and cytoskeleton; in cardiomyocytes, it is determined to exhibit as α1, α3, α5, α6, α7, α9, α10, β1, β3, and β5 subunits [42].

#### 4.1.6. Cadherins

In contrast to integrins, cadherins are composed of membrane proteins that enable the interaction of cells among themselves. When two cells share the same cadherins, they form a homophilic environment. Configurational changes in cadherins are hypothesized to be caused by calcium, which allows them to bind with other cadherins [43]. Cadherins interact with one other extracellularly and intracellularly in actin cytoskeletons. The intracellular interaction helps cadherin from cadherin-catenin complexes by further binding with the cytoskeleton, which induces MSCs cellular condensation, and eventually generates chondrogenic differentiation to occur [30,44]. Additionally, beta-catenin is known to have a role in several signaling pathways [45]. Some researchers believe OFF operates to deconstruct cadherin–catenin complexes, allowing b-catenin to serve as an osteogenic-stimulating molecular signaling molecule in MSCs [31,44]. *N*-cadherins, a cadherin subclass, has also been shown to be essential for MSC myogenesis [45]. A mounting body of work also exists pointing to a key role for the cadherins in the mechanotransduction of MSCs differentiation. 

#### 4.1.7. Stretch-Activated Ion Channels (SACs)

Stretch-activated ion channels (SACs) are known to be found in the plasma membrane of various types of cells, which have been shown to influence MSC differentiation with the aid of mechanical signals [31]. It can be explained through the transmission of mechanical tension from induced stress fiber formation and ion channels activation, whereas a subsequent study revealed that inhibiting ROCK resulted in less sensitivity of ion channels. Additionally, the sensitivity of ion channels can be controlled through Rho-dependent actin remodeling [38]. Further inhibition of ion channels is shown to subdue myogenesis, whereas numerous studies suggest ionic concentration greatly impacts myogenic differentiation and induces mechanotransduction [29]. This ionic concentration is influenced by tension caused by stress fiber and ion channel sensitivity. The mechanical signal transmitted upon mechanotransduction encourages SACs activation, subsequently inducing high ion transients in cells that cause cardiac electrical activity, which leads to depolarization in the membrane, and formation of the cytoskeleton as well as provides a long action potential duration [33]. It has been demonstrated that cardiomyocyte contractility occurs through the interchange of intracellular space and the influx of Na^+^ and Ca^2+^ ions. In addition, streptomycin and gadolinium, known as SAC blockers, can block the intracellular accumulation of Ca^2+^ ions, while the increase in intracellular Na^+^ levels simultaneously elevates intracellular Ca^2+^ via Na^+^ and Ca^2+^ exchange, which generates rapid contraction in cardiomyocytes [33].

### 4.2. Intrinsic Substrate Mechanics

The existence of cells in tissues controls the tissue’s stiffness, particularly from soft brain tissue to hard cortical bone. In vitro, it has been shown that substrate stiffness impacts MSCs lineage differentiation [3]. Cellular morphology, transcript indicators, and protein synthesis all showed that MSCs acquired a phenotype that corresponded to tissue stiffness when they were grown on two-dimensional (2D) substrates mimicking neurogenic, myogenic, and osteogenic settings [1]. Current experiments conducted revealed that MSCs are more capable of producing greater adipogenic and chondrogenic effects when grown on soft substrates, whereas those grown on stiffer substrates are more capable of producing a stronger myogenic potential involving muscle [46]. For example, in 2D culture methods, stiffness is often shown to alter cell morphology, but MSCs in three-dimensional (3D) hydrogels have been demonstrated to preserve their spherical shape regardless of the hydrogel’s stiffness [47]. Yet, the destiny of encapsulated MSCs is still largely dependent upon the hydrogel’s rigidity, with stiffer gels favoring osteogenesis and softer gels supporting adipogenesis, according to Huebsch et al. [46]. Integrin binding is required for osteogenic differentiation on rigid substrates, but it has little or no influence on MSCs differentiation down adipogenic or neurogenic lines when grown on soft substrates [48].

When comparing the attachment and differentiation of cells on solvent casting substrates to those on electrospun substrates, the adhesion and differentiation of cells on solvent casting substrates were noticeably stronger [49]. Stiff substrates stimulated bone cell differentiation, whereas soft substrates enhanced cell proliferation and migration. The degree of stiffness of the substrate has a direct impact on cell adherence, and this, in turn, has a significant impact on cell spreading. As a result, cell adhesion should be intermediate to allow for adherence and adaptation to the substrate while at the same time allowing adequate cell freedom for division and communication with other cells; adhesion, as a result, has an indirect effect on cell viability and survival. Stiffness impacts both differentiation and proliferation, and a rise in the ratio of these two measures show that the cell population is metabolizing properly. Differentiation is induced by stiffness, whereas adhesion is induced by suitable roughness [50].

In summary, the stiffness of a substrate influences a variety of cell processes, including cell shape, adhesion, migration, differentiation, and proliferation. Cell morphology has recently garnered renewed interest, owing to the possibility that measuring, predicting, and regulating cellular form would be useful in future tissue regeneration. External environmental and biomechanical stimuli alter the morphology of MSCs, which is described by the cell’s ability to balance exterior biomechanical stresses with internal forces in response to those forces. Thus, the stiffness of the biomaterial substrate is directly related to the exerted internal forces.

#### 4.2.1. Substrate Stiffness

While increasing matrix stiffness in both 2D and 3D substrates, the number of integrins attached to the matrix creates an exponential bell curve distribution, with hydrogel stiffness that permits peak bond formation, which allows optimum rigidity for osteogenic differentiation [44]. Myosin contractile machinery inhibition reduced the cell’s dependency on matrix stiffness for connection formation, showing that traction-mediated forces are required for the cell to assess its mechanical cues [51] appropriately. MSC lineage commitment was diminished by inhibiting link formation, demonstrating that matrix stiffness directly impacts MSCs lineage differentiation in cells [47]. MSCs-based myosin contraction is acquired to overcome substrate stiffness in 2D. Substrate stiffness may influence cell shape, especially in 2D cultures, owing to substrate stiffness-mediated changes in integrin binding, adhesion strength, and cell contractility, as illustrated in Figure 3b. The cytoskeleton’s internal arrangement and interactions with the extracellular matrix (ECM) involving neighboring cells affect the formation of a cell. For these reasons, the observed benefits of MSC differentiation have been demonstrated to be strongly influenced by cell shape [29].

#### 4.2.2. Cell Shape

A study by Steward and Kelly [30] showed that cell shape influences MSC development through seeding cells on micropatterned fibronectin-coated islands of varied sizes and then stimulating the cells with a mixed medium that allows for numerous lineages of differentiation. Small islands had a higher proportion of adipogenic colonies as compared to bigger islands, which had more of an osteogenic colony to determine MSC morphology [44]. This research concluded that cell shape is widely influenced by RhoA GTPase (RhoA) and Rho-associated protein kinase (ROCK) activity. In myosin contraction, ROCK is a Rho effector that regulates contractility. Cells exposed to the adipogenic medium developed an osteogenic phenotype when ROCK was inhibited; it was in contrast when RhoA was activated in osteogenic cells. This suggests that cellular contractility regulates MSC lineage commitment to either an osteogenic or adipogenic pathway [30,31,44]. TGF-beta 3-stimulated MSCs were either permitted to flatten and disseminate, or to retain their spherical cell shape, as per incomparable research. A myogenic lineage developed in MSCs that were allowed to expand, whereas a chondrogenic lineage developed in those that were maintained rounded [52]. Rac1, known as the Rho GTPase family, was elevated in the spread cells, but it was sufficient to drive myogenesis and inhibit chondrogenesis. However, TGF-b3 and Rac1 were discovered to upregulate a cell-to-cell adhesion molecule, showing that structural alterations to the cytoskeleton play a critical role in defining MSC lineage commitment through numerous routes [31].

### 4.3. Extrinsic Mechanotransduction Cues

Extrinsic mechanical cues, such as fluid flow, hydrostatic pressure, compression, and tensile stress, including the biological and mechanical features of the surrounding matrix, are all taken into account by mechanotransduction in the differentiation of MSCs indicated in Figure 3c [30,44]. MSC differentiation is influenced by various factors, including the type, frequency, amplitude, and length of these signals. To better understand how MSCs respond to diverse types of external mechanical impulses, the next sections will focus on the cellular response to these signals.

#### 4.3.1. Fluid Flow

Oscillatory fluid flow (OFF) has been shown to control stem cell fate, especially when degraded in the bone, as shown in vivo studies [29]. Actin fibril density, Rho and ROCK signaling, and expression of Runx2, Sox9, and PPARc in murine C3H10T1/2 progenitor cells have been enhanced by OFF according to Woods, Wang, and Beier [37]. When ROCK, NMM II, and actin polymerization inhibitors were added to fluid flow, there was no pro-osteogenic reaction. In the case of adipogenesis and chondrogenesis, it was shown that reducing cytoskeletal tension increased the expression of Sox9 and PPARc while further exacerbated by any fluid flow effects [31]. MSCs lineage pathways are affected by an intact, dynamic actin cytoskeleton, and most importantly, it is necessary for the transmission of OFF. Fluid flow stimulation has also been shown to promote the expression of osteogenic markers in MSCs generated from adipose tissue and bone marrow. MSCs differentiation induces the activation of mitogen-activated protein (MAP) kinases, such as ERK1/2, which have been suggested to increase intracellular calcium ions and additionally activate mitogen-activated protein (MAP) kinases [26].

Bone marrow-derived MSCs that have undergone cardiomyogenic differentiation could increase direct contributions from increased fluid flow [38]. Several studies revealed that primary cilia play an important role in mechanotransduction, which involves OFF [30]. Another key role of fluid flow is regulating MSCs’ fate in 3D constructions. MSC osteogenesis has been shown repeatedly to be aided by perfusion systems [14]. It has been previously shown that various forms of fluid flow have distinct effects on osteogenesis in a range of scaffolds [31]. Perfusion flow may cause mechanical cues in the cell, such forces could subsequently increase the overall nutrition and gas transport through the gel, making it difficult to interpret the outcome. Additionally, computational models are being extensively used in recent applications to examine the complicated interactions between fluid flow, cells, and the surrounding substrate [25].

#### 4.3.2. Hydrostatic Pressure

MSCs differentiation holds great promise for cell expression and is greatly influenced by hydrostatic pressure (HP). Numerous studies have indicated that cyclic HP promotes Sox9 [38], aggrecan, and collagen type II chondrogenic gene expression as well the production of proteoglycans [44] and collagen in MSCs [53]. However, several studies reported the limitation showing that HP has no substantial impact on MSC chondrogenesis due to the encapsulated cells maintaining the response toward HP and relying on the substance within the cells [2,30]. Hence, interactions between cell and substrate may influence MSCs response towards HP, which acts as external mechanical stimuli in evaluating MSCs fate. In addition, hydrostatic pressure involves a mechanical stimulus that does not deform, thus, making it difficult to identify possible mechanotransduction pathways. The effect of intracellular calcium concentration induces a possible mechanotransduction signal for chondrocytes involving HP [31]. Na/H exchange in chondrocytes is enhanced by inhibiting the Na/K pump and the Na/K/2Cl pump. Intracellular calcium stores provide an evident increase in chondrocyte intracellular calcium concentration, which has been identified in the 30s of static HP.

Recent studies have determined that the cytoskeleton is affected by hydrostatic pressure. In epithelial cells, increased magnitude static HP prevents the production of both microtubules and actin fibers, resulting in cell rounding [31]. However, undamaged dynamic microtubules were acquired for the mechanotransduction of cyclic high-magnitude static HP by chondrocytes; however, the disrupted microtubules mainly contributed by low matrix synthesis in chondrocytes [53]. It was shown that agarose hydrogels offered a higher pro-chondrogenic environment; meanwhile, fibrin hydrogels had a more robust response to the administration of HP. Agarose hydrogels were seeded with MSCs in either a soft or stiff form; later investigations revealed cytoskeletal organization and formation of focal adhesions were different in stiff gels than in soft gels, showing MSCs chondrogenic phenotype was more prominent in the soft gels. However, only the stiffer gels responded positively to HP treatment with a pro-chondrogenic phenotype [31]. Meanwhile, cell attachment and stress fiber production may inhibit chondrogenesis, which induces mechanotransduction of HP in MSCs. HP has been demonstrated to modify vimentin structure, indicating a new function for intermediate filaments in the mechanotransduction of HP.

#### 4.3.3. Compression

MSCs, together with 3D hydrogels, were directly compressed to provide a significant pro-chondrogenic stimulation [2]. Subsequently, dynamic compression allows chondrogenic gene expression in MSCs and inhibits exogenous growth factor stimulation, resulting in chondrogenesis [31]. Numerous studies have demonstrated that compression and exogenous transforming growth factor-β1 (TGF-β) stimulus activate identical pathways that lead to increases in (TGF-β1) gene expression via compression [26]. The effects of activating MSCs with soluble (TGF-β1) and Dynamic compression (DC) are still in progress, even though this process induces chondrogenesis. However, it has been revealed that the combined administration of (TGF-β3) and mechanical stimulation may suppress chondrogenesis. Chondrogenesis in MSCs may be enhanced through-loading after extended exposure to (TGF-β1) [31].

#### 4.3.4. Tension

The tensile strain has been implicated as a key regulator in expressing fibrogenic [54], osteogenic [44], and chondrogenic [30,39,52] in MSCs markers. Cyclic tensile strain exerted in MSCs could increase BMP-2 expression, osteogenic gene expression, and calcium deposition in MSCs. Further, it has been found that MSCs were subjected to cyclic tensile increases in the MAP kinase pathway, which is a key mechanotransduction route in osteogenic differentiation [31].

Some studies revealed that ERK and p38 were needed for mechanotransduction of cyclic tension to exhibit high collagen I gene expression involving tension besides subsequently activating the stretch-activated cation channels [54]. In contrast, other studies have demonstrated that tension has no significant effect on chondrogenic-related genes, which might be explained, at least in part, by the finding that the response to tension may depend on the osteogenic and fibroblastic genes during compression exhibit enhancement in many chondrogenic-related genes [55]. Most b-catenin, according to Haudenschild et al. [56], suggested being upregulated by dynamic tension to maintain the interaction between cells and subsequently inhibit chondrogenesis [26,57].

Overall, an intact and dynamic tension may regulate osteogenic response due to cell-cell interaction. It has been possible to determine the role of cyclic tensile on MSCs directly, studies suggesting MSC were seeded into collagen-GAG scaffolds, subjected to cyclic tensile strain stimulating more proteoglycans than unstrained controls, revealing that this application might cause the release of tensile signal, which eventually may lead to a pro-chondrogenic response [30,39]. In addition, the magnitude of cyclic tension has also been found to promote MSC fate; high cyclic tension favors myogenesis, whereas a low strain is more favorable to osteogenesis in the absence of growth stimuli for rabbits seeded with MSCs [50,54]. The cyclic tensile strain has consistently been found to influence the expression of proinflammatory cytokines [58]. It has been established that mechanical stimulation not only stimulates osteogenesis but it also helps to sustain bone formation [29].

## 5. How Mechanotransduction Changes the Clinical Application of MSC

Following mechanotransduction, MSCs have been shown to differentiate into their possible lineages, which can ultimately change the prospect of MSCs in a clinical setting. Mechanical forces can determine the transition of inflammation from systemic autoimmunity to local inflammation. In 2018, Cambré et al. revealed that MSCs in the mechanically sensitive areas of joints sense mechanical stimuli and convert mechanical stimuli into chemical stimuli, causing local inflammation and bone destruction, eventually leading to the development of arthritis [59]. Thus, mechanical forces play an important role in the local inflammatory response in the human body [2]. The acute inflammation process begins with neutrophil activation, which causes inflammatory proteins and chemokines to be secreted to attract monocytes and macrophages [60]. In addition to removing necrotic tissue, macrophages secrete inflammatory cytokines and chemokines such as TNF-α, IL-1β, IL-6, and CCL2 to mobilize MSCs to replace hematoma. Shortly thereafter, MSCs are stimulated by various factors in the environment, inducing bone formation and differentiation via either endochondral ossification or intramembranous ossification. Therefore, it is clear that the proper duration of the acute inflammatory phase is important for bone regeneration. However, excessive inflammatory reactions often result in failure of bone repair in bone tissue engineering. Several studies have shown that mechanical stimuli may promote the elimination of inflammation by regulating the interaction between MSCs and macrophages. Therefore, investigations of anti-inflammatory mechanisms and optimal application parameters are of great importance [61].

Zhang et al. [62] created a hydroxyapatite scaffold made from the extracellular matrix of compression stimulated MSCs by freeze-drying the ECM. This bio-fabricated scaffold has the potential to speed macrophage polarization from the pro-inflammatory M1 towards the anti-inflammatory M2 phenotype, hence promoting bone repair [63]. These data imply that compression may stimulate the release of anti-inflammatory molecules in MSCs. Recent research, however, has discovered that MSCs maintain physiological levels via TNF-α endocytosis [64]. TNF-α endocytosis (via cyclic stretching) boosted MSCs proliferation and osteogenic differentiation via downregulating TNF-α secretion in MSCs rather than directly downregulating TNF-α gene expression [64]. Mechanical stimulation of adipose tissue also influences the anti-inflammatory capabilities of human adipose-derived stem cells (hADSCs) in adipose tissue. Carelli et al. [65] examined the anti-inflammatory characteristics of human adipose-derived stem cells (hADSCs) in physically stimulated adipose tissue compared to those of the control group. They discovered that mechanically stimulated-hADSCs outperformed control groups (hADSCs in terms of anti-inflammatory activity) [66]. Other investigations, however, have discovered that mechanical stimulation can induce both inflammation and osteogenesis at the same time, most likely due to the MSC’s autocrine control of inflammatory factor production [67,68].

According to Dong et al. [69], macrophages can respond to mechanical stimulation as well. Mechanical strain causes macrophages to polarize into the M2 phenotype, which secretes inflammation-related cytokines, including IL10 and TGF- to control the local inflammatory milieu. Mechanical stimulation stimulates macrophages’ YAP/BMP2 axis, resulting in increased BMP2 expression and MSC osteogenesis. YAP, an important element of the mechanical signaling pathway, causes the polarization of M2 macrophages via Wnt5a and TGF1 [61]. Schoenenberger et al. [70] discovered that mechanosensitive macrophages play an important role in tendon healing. A few studies have shown that mechanical stimulation can help with inflammation resolution by modulating the interactions of MSCs and macrophages. In orthopedics, mechanical stimulation, also known as mechanotherapy, has been employed as a treatment [62]. Distraction osteogenesis, for example, is used to rectify limb and craniofacial deformities, whereas low-intensity pulsed ultrasound (LIPUS) is utilized to accelerate fracture healing and enhance bone mass. On the other hand, the current technique uses mechanical stimulation directly on the tissue rather than via the substrate. However, the effectiveness of these mechanotherapy in bone healing is debatable, with a recent comprehensive study concluding that LIPUS did not enhance significant clinical outcomes. Mechanical stimulation-induced macrophage M2 phenotypic transition and subsequent tissue repair [61]. These findings imply that further research into the effect of mechanical stimulation in MSC and macrophage co-culture models should be considered in the future.

Overall, the regulation of mechanical stimulation on MSC presented in various studies is of considerable importance for regenerative medicine, as it provides insight for the discovery of mechanosensors, which not only elucidates the processes of mechanotransduction and fate determination but also opens up new avenues for mechanical control in clinical applications. Furthermore, the portrayal of various mechanical stimulation application modalities provides informative assistance in engineering regenerative biomaterials and clinical uses of mechanical stimulation.

## 6. Future Perspectives

In 2015, Steward and Kelly [30] reported that most experiments were conducted to study the mechanobiology of MSCs involved in vitro, while in vivo, mainly based on the observations made upon the hypothesis that dependent on mechanical signaling and their development relies on the repair of musculoskeletal tissues. Therefore, in vitro modeling provides improvisation of understanding MSCs respond to specific stimuli compared to in vivo, which involves both intrinsic and extrinsic mechanical cues beside the biophysical and biochemical environment, which is considered inherently more complex. The mechanotransduction mechanisms investigated through isolating single variables via computational models ease the understanding of the interplay of several different factors simultaneously, in addition to various environmental factors inducing MSC fate in vivo [71]. Besides the advanced tools used in determining MSC’s fate, the investigation involving the interaction among mechanical cues and soluble signals may advance bioengineering technologies allowing the development to craft complex tissues and organs for regenerative medicine applications [9]. This may induce MSCs therapy to play a vital role in treating diseases. Stem cell therapy involving MSCs such as umbilical cord mesenchymal stem cells (UC-MSCs) offers multiple differentiation capabilities, easy access, immune exemption and ethical requirements, and large-scale expansion, which most importantly synthesizes bioengineered medicine for IHD diseases [60,61]. However, mechanotransduction in pathological conditions that occur when viruses infect MSCs might be different.

On the other hand, iPSCs obtained from patients show no immune rejection upon transplantation and have the characteristics of differentiation into a variety of cells, organoids under the induction of small molecule compounds, which are highly expressed in screening as drugs and evaluated based on their efficacy [72]. Most studies that explored cardiac tissue engineering suggested generation of thick vascularized tissues that fully match the patient is still in progress. However, Noor et al. [23] successfully constructed a cellularized human heart with major blood vessels from endothelial cell-laden hydrogels and cardiac cells using 3D bioprinting, and such print has improved our understanding of engineering personalized tissues and organs for future use [73]. According to Rinoldi et al. [74], MSCs respond to mechanical and biochemical stimuli, improve the efficiency of tissue engineering and serve as useful tools to investigate the current stem cell therapy. MSCs, tissue engineering, and 3D printing technology, as promising methods of tissue regeneration and organ transplantation, will receive more and more attention. Advancement in imaging allows minimally invasive and even non-invasive cell transplantation, including cell tracing via multimodal imaging and ease of MSCs mechanism [75].

## 7. Conclusions

The recent interpretation involves a mechanism that interconnects stem cells and their microenvironment wherein mechanical stresses are applied, driving critical stem cells’ fate, including MSCs. Although much research is still ongoing, it is already clear that the mechanical environment of MSCs is determined by ECM, with numerous membrane proteins, cytoskeletal components, and the nucleus itself all acting as putative mechanosensors could be translated to the cell through mechanosensing or mechanotransduction signals that elicit adaptive responses. The complex interactions between these diverse actors represent a general scheme by which cells, tissues, organs, and whole organisms respond to external and internal mechanical stimuli orchestrating their biological activity even if they are still not fully understood. Further research is needed to elucidate how MSCs sense and respond to the complex sets of mechanical stimuli, whether in vivo or in vitro, followed by their development in regenerative tissue engineering.

## Figures and Tables

**Figure 1 ijms-23-04580-f001:**
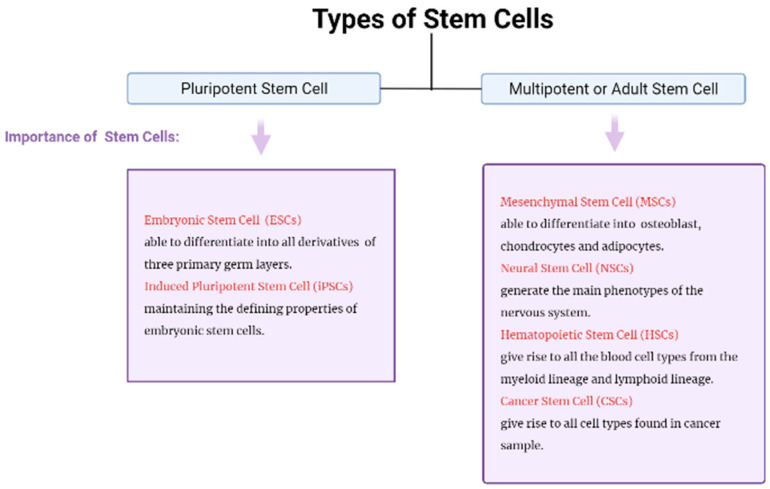
The types of stem cells and their importance.

**Figure 2 ijms-23-04580-f002:**
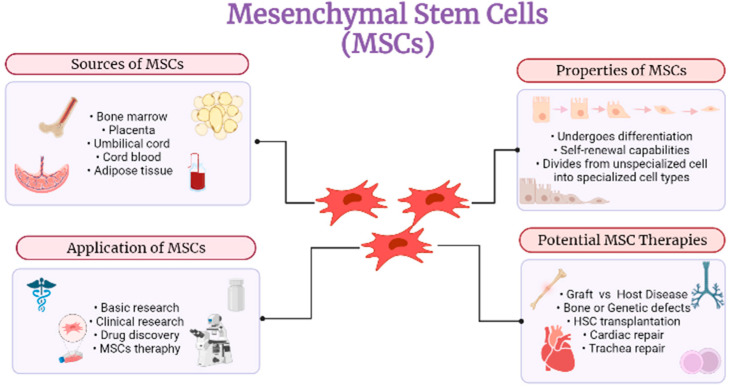
The overview of Mesenchymal Stem Cells (MSCs).

**Figure 3 ijms-23-04580-f003:**
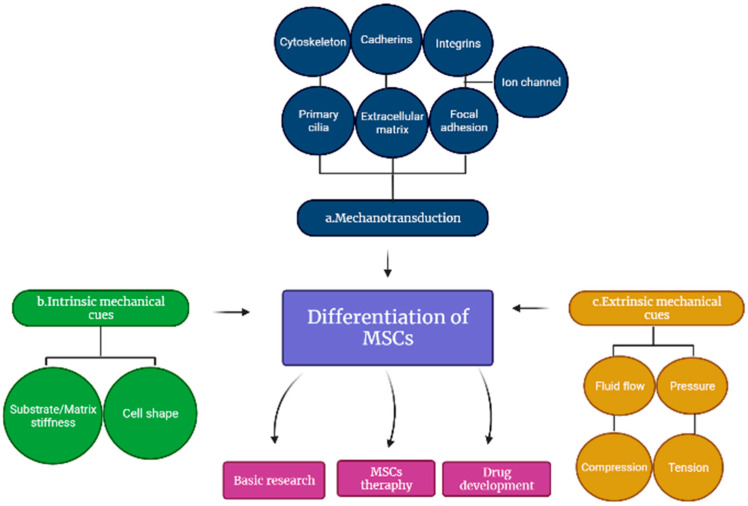
Differentiation of mesenchymal stem cells (MSCs). The differentiation involves (**a**) cell membrane and numerous intracellular components, (**b**) intrinsic and (**c**) extrinsic mechanical cues that regulate the mechanotransduction of MSCs.

## Data Availability

Not applicable.

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
