# Peer review of "Mechanotransduction in Mesenchymal Stem Cells (MSCs) Differentiation: A Review"

_ijms, 2022, doi:10.3390/ijms23094580_

Round 1

Reviewer 1 Report

The manuscript is a review article on the mechanotransduction in differentiation of MSCs. While the aim of the study is not clearly stated. This is generally accepted importance of mechanotrasduction to control properties of MSCs. Numerous reviews on the subject have been published already. Considering this, the topic of the current review looks interesting and useful but challenging. In light of the available literature, it would be reasonable to expect seeing a mature, thought-provoking or visionary review under the title mentioned above. However, and very unfortunately, the presented manuscript has a limited scope, shallow depth and rather listing a random fact than providing the analysis of the current state of the field.

The overview of the mechanical influences in the differentiation MSCs is significantly incomplete and, sometimes, it’s misleading. In the review of Mechnotransduction in MSCs differentiation as substrate stiffness and extrinsic mechanotransduction cues, only a very superficial presentation on the subject is given. Importantly, there are no signs of analysis of state-of-the-art methods and new ideas in the field. In contrast, there is mostly a repetitive listing of common pieces of knowledge.

Regrettably, I am recommending the rejection of the manuscript in the current state as its’ quality if not satisfying the criteria of Q1 journal publication.

Author Response

Thank you for your suggestion. We have highlighted the significance of the review in the abstract portion of the paper, we believe that the review has been structurally useful and easy to comprehend (page 1, lines 33-37).

Reviewer 2 Report

Please add more to the topic of mechano-electric coupling as well as to the topic of height and changes of the resting membrane potential in MSCs. Because it is known that the level of the resting membrane potential can influence function and differentiation of MSCs and their cell cycle phase. 

Author Response

Thank you for your comment and suggestion. However, because the study itself is too lengthy to include the part on mechano-electro coupling, and because we didn't include electrochemical signalling in our review, we decided to leave it out. We did draw attention to prior research concerning the level of the resting membrane potential since it has the ability to impact the function and differentiation of MSCs as well as their cell cycle phase under the (3) functions of MSCs heading (page 5, lines 172-180).

Reviewer 3 Report

Comments to Authors

The authors addressed the effects of mechanotransduction on MSC differentiation by categorizing mechanical forces into three major chapters: cell membrane and intracellular components, intrinsic substrate mechanics, and extrinsic mechanotransduction cues.

The originality of this paper stems from the freshness of this field of investigation, which is still relatively new, and so there is a limited data of articles on the subject.

Major comments:

The authors focused on the effect of mechanotransduction on MSC differentiation. Given that substrate stiffness has been identified as a significant factor in cell spreading, migration, proliferation, and differentiation in recent studies, it would be interesting if the authors mentioned this important interrelationship between the multiple factors.

The role of biophysical and mechanical stimulation in the biogenesis of extracellular vesicles has been demonstrated in several studies. "Mechanobiology" is the study of the relationship between biophysical/mechanical stimulation and EV biogenesis. It would be interesting if the authors discussed about the significance of this signaling pathway.

It would be interesting if the authors included an analysis of mechanotransduction in pathological conditions that occur when viruses infect MSCs, at least in the future perspectives section. Viruses infecting MSC use and alter cell membrane and intracellular components involved in mechanotransduction during uptake, replication, and exit from the cell.

Author Response

In response to the reviewer's comment, we have added that substrate stiffness has been identified as a significant factor in cell spreading, migration, proliferation, and differentiation in recent studies, and that it has been mentioned about its important interrelationship between the multiple factors under the subheading of (4.2). Intrinsic substrate mechanics (page 10, lines 380-399).

Thank you for your suggestion. However, EV biogenesis is not discussed at all in this review, since the study itself is too lengthy to include the part. However, we will consider to write another topic in future related to EV biogenesis.

In regards to the third suggestion, viruses are beyond the subject of this article; however, we did mention MSCs therapy employed against the COVID-19 under the future perspective section (page 13, lines 564-571).

Round 2

Reviewer 1 Report

-

Author Response

Dear reviewer, 

We have proofread the manuscript and hope it would be able to be accepted soon. Thank you.

Reviewer 2 Report

Now, an interesting picture emerges about the story of mechanotransduction and mesenchymal stem cells.

Author Response

Dear reviewer,

Thank you for your positive review. We have proofread the manuscript and hope it would be able to be accepted soon.

Reviewer 3 Report

Comments to the Author
In the revised manuscript, the authors have responded appropriately to the suggestions, offered more information, and successfully discussed some of the article's problematic aspects, for which I think improves the quality of the manuscript a lot. I agree to accept this manuscript.

Author Response

Dear reviewer,

Thank you for your positive feedback. We hope this manuscript will be accepted soon.
